# A Second Gamma-Glutamylpolyamine Synthetase, GlnA2, Is Involved in Polyamine Catabolism in *Streptomyces coelicolor*

**DOI:** 10.3390/ijms23073752

**Published:** 2022-03-29

**Authors:** Sergii Krysenko, Nicole Okoniewski, Merle Nentwich, Arne Matthews, Moritz Bäuerle, Alina Zinser, Tobias Busche, Andreas Kulik, Stephanie Gursch, Annika Kemeny, Agnieszka Bera, Wolfgang Wohlleben

**Affiliations:** 1Department of Microbiology and Biotechnology, Interfaculty Institute of Microbiology and Infection Medicine Tübingen (IMIT), University of Tübingen, Auf der Morgenstelle 28, 72076 Tübingen, Germany; sergii.krysenko@uni-tuebingen.de (S.K.); nicole.okoniewski@eawag.ch (N.O.); merle.nentwich@uni-tuebingen.de (M.N.); arne.matthews@zbsa.de (A.M.); moritz.baeuerle@student.uni-tuebingen.de (M.B.); alina.zinser@student.uni-tuebingen.de (A.Z.); andreas.kulik@uni-tuebingen.de (A.K.); sgursch@web.de (S.G.); annika.kemeny@uni-tuebingen.de (A.K.); agnieszka.bera@biotech.uni-tuebingen.de (A.B.); 2Cluster of Excellence ‘Controlling Microbes to Fight Infections’, University of Tübingen, 72076 Tübingen, Germany; 3EAWAG (Swiss Federal Institute of Aquatic Science and Technology), 8600 Dübendorf, Switzerland; 4Ampack GmbH, Bosch Packaging Technology, 86343 Königsbrunn, Germany; 5Faculty of Biology, University of Freiburg, 79098 Freiburg, Germany; 6Center for Biotechnology (CeBiTec), Bielefeld University, 33501 Bielefeld, Germany; tbusche@iit-biotech.de; 7Vetter Pharma-Fertigung GmbH & Co. KG, 88212 Ravensburg, Germany; 8Department of Microbial Active Compounds, Interfaculty Institute of Microbiology and Infection Medicine Tübingen (IMIT), University of Tübingen, Auf der Morgenstelle 28, 72076 Tübingen, Germany

**Keywords:** nitrogen assimilation, GS-like enzymes, GlnA, GlnA2, GlnA3, polyamine catabolism, *Streptomyces coelicolor*

## Abstract

*Streptomyces coelicolor* is a soil bacterium living in a habitat with very changeable nutrient availability. This organism possesses a complex nitrogen metabolism and is able to utilize the polyamines putrescine, cadaverine, spermidine, and spermine and the monoamine ethanolamine. We demonstrated that GlnA2 (SCO2241) facilitates *S. coelicolor* to survive under high toxic polyamine concentrations. GlnA2 is a gamma-glutamylpolyamine synthetase, an enzyme catalyzing the first step in polyamine catabolism. The role of GlnA2 was confirmed in phenotypical studies with a *glnA2* deletion mutant as well as in transcriptional and biochemical analyses. Among all GS-like enzymes in *S. coelicolor*, GlnA2 possesses the highest specificity towards short-chain polyamines (putrescine and cadaverine), while its functional homolog GlnA3 (SCO6962) prefers long-chain polyamines (spermidine and spermine) and GlnA4 (SCO1613) accepts only monoamines. The genome-wide RNAseq analysis in the presence of the polyamines putrescine, cadaverine, spermidine, or spermine revealed indication of the occurrence of different routes for polyamine catabolism in *S. coelicolor* involving GlnA2 and GlnA3. Furthermore, GlnA2 and GlnA3 are differently regulated. From our results, we can propose a complemented model of polyamine catabolism in *S. coelicolor*, which involves the gamma-glutamylation pathway as well as other alternative utilization pathways.

## 1. Introduction

The Gram-positive, non-motile soil bacterium *Streptomyces coelicolor* belongs to the genus *Streptomyces*, phylum Actinobacteria, which is well known for its ability to produce secondary metabolites. This organism features a complex nitrogen metabolism required for the survival of *S. coelicolor* under changeable nitrogen conditions in soil. *S. coelicolor* is able to assimilate nitrogen from diverse mineral (e.g., ammonium, nitrate, and nitrite) and organic sources (e.g., urea, amino acids, peptides, amino sugars, polyamines, and monoamines). Previous work on polyamine metabolism revealed that the glutamine synthetase-like enzyme GlnA3 makes possible the survival of *S. coelicolor* under extreme environmental stress conditions of local polyamine exposure [1,2].

Polyamines are positively charged aliphatic polycations with a polycarbon chain and multiple amino groups. Polyamines are present in organisms of all kingdoms of life. Widely distributed natural poylamines include putrescine, cadaverine, spermidine, and spermine, but there are also more long-chain compounds reported in bacteria and archaea [3]. Polyamines were demonstrated to be essential for the growth of bacteria [4]. Additionally, these molecules are involved in cellular processes such as regulation of transcription and translation, cell growth stimulation and biofilm formation, stress resistance, and biosynthesis of siderophores [3,5]. Furthermore, polyamines can modulate the biosynthesis of secondary metabolites. It has been demonstrated that putrescine can induce the secondary metabolism in *Nocardia lactamdurans* [6]. Spermidine has been shown to stimulate the biosynthesis of benzylpenicillin in *Penicillum chrysogenum* and produce a drastic increase in the transcript levels of the penicillin biosynthetic genes [7]. In contrast with putrescine and spermidine, the role of cadaverine and spermine in bacteria remains not well studied.

Polyamines can be synthesized *de novo,* being predominantly derived from the amino acids ornithine, arginine, and lysine, which can be decarboxylated to form cadaverine or putrescine, a precursor of spermidin and spermin [8]. However, their concentrations and presence are very variable between species: e.g., in *E. coli*, 10–30 mM of intracellular putrescine was reported, while in most bacteria the putrescine amount ranges between 0.1 and 0.2 mM and spermidine between 1 and 7 mM [9,10]. In *S. coelicolor*, the biosynthesis of putrescine and spermidine has been reported to occur in the late-stationary phase, whereas cadaverine synthesis happens only under iron limitation [11].

Control of the intracellular polyamine level is crucial for regulation of their intracellular pool. Polyamine utilization allows the detoxification of high polyamine concentrations, which are toxic for cells, and the use of them as an alternative N-source. For this, diverse routes for polyamine catabolism are present in bacteria. The currently known polyamine degradation pathways in prokaryotes include the gamma-glutamylation pathway, the aminotransferase pathway, the spermine/spermidine dehydrogenase pathway, the direct oxidation pathway, and the acetylation pathway (Figure 1) [2,12,13,14,15].

Aminotransferase pathway has been described in *Escherichia coli* and involves an aminotransferase (Figure 1, AMTP) [13]. The gamma-glutamylation pathway has been studied in *E. coli* [13,16], *Pseudomonas aeruginosa* [12], and *Streptomyces coelicolor* [1], involving gamma-glutamylpolyamine synthetase as a key enzyme (Figure 1, GGP). The direct oxidation pathway has been investigated in *P. aeruginosa* and requires an amine oxidase (Figure 1, DOP) [12]. The acetylation pathway has been reported in *E. coli* and *B. subtilis*; it involves an acetyltransferase (Figure 1, ACP) [12,14,15,17]. The spermine/spermidine dehydrogenase pathway has been described in *P. aeruginosa* [12], for which the structure of the key enzyme spermidine dehydrogenase has been reported (Figure 1, SPDP) [18].

**Figure 1 ijms-23-03752-f001:**
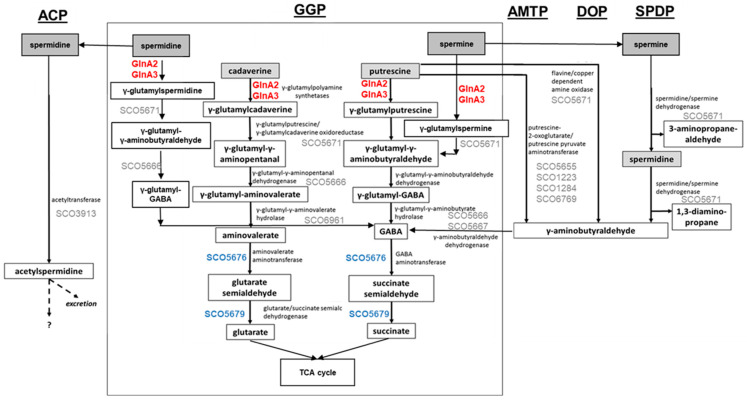
Combined model of polyamine utilization pathways within prokaryotes and in *S. coelicolor* based on published studies [2,4,12,13,14,15,17]. GGP, gamma-glutamylation pathway; AMTP, aminotransferase pathway; DOP, direct oxidation pathway; SPDP, spermine/spermidine dehydrogenase pathway; ACP, acetylation pathway. Dashed arrows represent predicted metabolic pathways. Enzymes in red: confirmed gamma-glutamylpolyamine synthetases in *S. coelicolor*. Proteins in blue: confirmed role in polyamine utilization in *S. coelicolor*. Proteins in grey: predicted role in polyamine utilization in *S. coelicolor*.

All pathways aim for the neutralization of polyamines. However, some of these pathways yield toxic intermediates or end products: gamma-aminobutyraldehyde appears in the aminotransferase (Figure 1, AMTP), direct oxidation (Figure 1, DOP), and spermine/spermidine dehydrogenase (Figure 1, SPDP) pathways. Acetylspermidine occurring in the acetylation pathway (Figure 1, ACP) may not be metabolized further and may need to be excreted. In contrast, the gamma-glutamylation pathway (Figure 1, GGP) is suitable for further metabolization of polyamines as N-sources, and it was demonstrated that in this pathway polyamines can be reduced to glutarate and succinate that enter the TCA cycle [4,19].

The enzyme catalyzing the initial step of the gamma-glutamylation pathway is gamma-glutamylpolyamine synthetase ligating polyamines with glutamate. In *E. coli*, PuuA has been reported to perform this reaction [20]. *P. aeruginosa* possesses an expanded polyamine catabolic pathway involving seven related gamma-glutamylpolyamine synthetases (PauA1-PauA7), which are all glutamylating enzymes with different substrate specificity towards polyamines, monoamines, or other substrates [12].

The polyamine gamma-glutamylation pathway was also described in *S. coelicolor*. The initial step in this pathway is catalyzed by the gamma-glutamylpolyamine synthetase GlnA3, which is able to glutamylate putrescine, cadaverine, spermidine, and spermine [1]. GlnA3 is one of three glutamine synthetase-like (GS-like) enzymes in *S. coelicolor*, which include GlnA2 (SCO2241), GlnA3 (SCO6962), and GlnA4 (SCO1613). These enzymes do not exhibit glutamine synthesis activity [21] that is achieved by two functional glutamine synthetases (GSs): GS type I—GlnA (SCO2198) and GS type II—GlnII (SCO2210) [22,23,24]. GlnA4 has been shown to glutamylate monoamines with a high substrate specificity towards ethanolamine and was characterized as a gamma-glutamylethanolamid synthetase [25]. For GlnA2 a dual function—catalytic and regulatory—has been hypothesized [26].

The occurrence of GlnA2 has also been reported for *Mycobacteria* sp., where this enzyme shows about 69% amino acid sequence identity compared with GlnA2 in *S. coelicolor*. It has been shown that in *Mycobacterium bovis*, a small deletion in the *glnA2* gene resulted in a loss of virulence [27]. However, the primary enzymatic function of *glnA2* in *M. bovis* and *S. coelicolor* remained unknown. Recently, GlnA2 has been described as a putrescine-glutamate ligase (HFX_01688) in the halophilic archaea *Haloferax mediterranei*, suggesting a role of GlnA2 in polyamine metabolism [28].

In this work, we demonstrate the enzymatic activity of GlnA2 as a gamma-glutamylpolyamine synthetase and describe its role in polyamine utilization in *S. coelicolor*.

## 2. Results

### 2.1. The Sequence and the 3D Model Structure of GlnA2 S. coelicolor M145 Differs from That of Glutamine Synthetases but Resembles That of Glutamate-Polyamine Ligases

Three genes encoding GlnA-like enzymes are present in the *S. coelicolor* genome. The occurrence of these genes in actinobacteria and *Streptomyces sp.* was investigated in an in silico analysis of available sequenced genomes by BLASTn. While *glnA* can be found in virtually all bacteria, analysis of representatives of 24 genera belonging to the order *Actinomycetales* revealed the occurrence of a *glnA2* gene sequence in about 96%, *glnA3* in 29%, and *glnA4* in 54% of the genera.

To the best of our knowledge there are no reports on the function of *glnA2* in Actinomycetes. It is only known that *glnA2* does not code for an enzyme with GS activity: the *glnA2* gene cannot complement a double mutant of the glutamine synthetase genes *glnA* and *glnII* in *S. coelicolor* [1,21]. In order to find indications on GlnA2 function, the amino acid sequence of GlnA2 was compared with glutamine synthetases. The sequence alignment of GS and GS-like proteins from *S. coelicolor* and other bacteria by BLASTp revealed 18 highly conserved amino acid residues. For instance, GlnA*_St_* from *Salmonella typhimurum* and GlnA*_Sc_* from *S. coelicolor* contain identical conserved amino acids, whereas GS-like proteins of *S. coelicolor* have multiple substitutions of conserved amino acids. GlnA2 has 32% identity/48% similarity with GlnA*_Sc_*. In *S. coelicolor*, GlnA2 and GlnA3 lack some amino acids present in the active site of GlnA (Appendix A). Of the 18 amino acid residues present in the active site of GS and important for their activity (Ser53, Asn264), 2 are absent in GlnA2, while other conserved residues—D50, Tyr179, Glu327, and W397—are present (Appendix A).

For further detailed in silico analysis of proteins, comparative 3D modeling of GlnA2 with the crystal structure of described glutamine synthetases was performed. The aim of the analysis of the GlnA2 3D model structure was to identify the localization of the amino acid residues present in GlnA but absent in GlnA2. The 3D constellation of these residues would provide information about the putative function of GlnA2. The model structure of GlnA2 was generated by SWISS-MODEL. As a template protein, GSI from *S. typhimurium* (PDB: 1FPY) [29] was selected from a large variety of proteins. This protein template yielded the best GlnA2 3D model, as evaluated using the QMEAN scoring function, the MolProbility score, and a Ramachandran plot. The GlnA2 3D model consists of six sub-units organized in two rings, forming a dodecameric enzyme (Figure 2A). A structural comparison of the GlnA2 model with the crystal structure of GSI from *S. typhimurium* (Figure 2B) revealed conserved binding sites for glutamate, ATP, and metal ions Mg^2+^/Mn^2+^ in the active site of GlnA2. Moreover, it revealed that the amino acid substitutions in GlnA2 are localized in the region corresponding to the ammonium binding site in GlnA: Ser53 -> Glu55, Tyr179 -> Asp150, Asn264 -> Pro238 and Glu327 -> Glu310 (Figure 2C; Table 1).

Additionally, a GlnA3 model based on the structurally analyzed GSI from *S. typhimurium* (PDB: 1FPY) was generated. GlnA3 is a closely related to GlnA2 that ligates glutamate with polyamines [1]. The comparison of its 3D model structure with that of GSI GlnA from *S. typhimurium* revealed substitutions of amino acid residues in the ammonium binding site: Ser53 -> Ser76, Asn264 -> Asn260, and Glu327 -> Thr327 (Table 1).

In the GlnA2 and GlnA3 model structures, the localization of amino acid residues essential for the catalytic activity at the ammonium binding site differs from GSI GlnA. This suggests the possibility of the binding of substrates other than ammonium, e.g., polyamines (Figure 2C). The significant overall similarity of the GlnA2 model to the model structure of the gamma-glutamylpolyamine synthetase GlnA3 as well as the lack of the conserved residues for ammonium binding present in GlnA suggested that GlnA2 may function as a gamma-glutamylpolyamine synthetase catalyzing the polyamine glutamylation.

### 2.2. GlnA2 Accepts Polyamine Substrates and Shows Gamma-Glutamylpolyamine Synthetase Activity

In order to determine whether the predicted in silico polyamine glutamylation reaction can be catalyzed by GlnA2 in vitro, an LC/MS analysis was carried out. For this, His-GlnA2 protein was produced in an *E. coli* BL21 (DE3) strain and subsequently purified by nickel ion affinity chromatography. The reaction mix for the in vitro reaction was containing the purified His-GlnA2 enzyme, glutamate, the polyamine putrescine, cadaverine or spermidine, ATP, and Mg^2+^ and was incubated at 30 °C for 5 min. Only a glutamylated putrescine, cadaverine, or spermidine can be synthesized under these assay conditions. In the reaction mixture with putrescine, which is the most widespread polyamine in nature, LC/MS analysis revealed a peak for a compound with the mass 218.0 *m*/*z* (+MS), which corresponds to gamma-glutamylputrescine (Figure 3 and Appendix A). This result correlates with the analogous experiment carried out with His-GlnA3, which demonstrated the generation of the gamma-glutamylputrescine [1]. Furthermore, in the reaction mixtures with cadaverine or spermidine, LC/MS analysis revealed a peak for a reaction product with the mass 231 *m*/*z* (−MS) and 274 *m*/*z* (−MS), respectively, which correspond to gamma-glutamylcadaverine and gamma-glutamylspermidine (Appendix A). However, the evaluation of these data was hindered due to the absence of the commercially available glutamylated polyamines that could be used as controls. To verify observed effects and to deepen the biochemical study of GlnA2, we applied an additional in vitro assay.

In order to study the enzymatic activity and substrate specificity of GlnA2, an adapted GS activity assay [30] based on the inorganic phosphate (Pi) released from ATP was used. An amount of 10 µg of His-GlnA2 enzyme was incubated with different polyamine substrates at a concentration of 50 mM. This assay was designed and optimized for rapid, high-throughput screening of different substrates in the microtiter format, allowing performance of multiple sets of kinetic experiments in parallel. The amount of released inorganic phosphate was quantified and expressed in nanomoles of Pi. The biochemical analysis of the GlnA2 activity and its substrate specificity using different polyamines revealed that GlnA2 can accept different polyamines as substrates (Figure 4 and Appendix A).

Interestingly, GlnA2 also demonstrated activity with ammonium. However, the ammonium concentration in the assay did not correspond to the physiological one. The substrate concentration in the assays was adjusted to the polyamine concentration reported for *S. coelicolor* growing in the presence of polyamines [1], which are about 50-fold higher than the physiological ones for ammonium. In in vivo tests GlnA2 did not show any GS activity [1,21]. The measured GS activity could presumably be attributed to a spontaneous enzyme reaction due to a conformational change in the enzyme associated with the excess of substrate, a process that was reported for a variety of other proteins [31].

For comparing the enzymatic activity of GlnA2 and GlnA3 in the presence of polyamine substrates, 10 µg of His-GlnA3 was tested as described above. The assay revealed activity of both GlnA2 and GlnA3 with polyamine substrates. However, GlnA3 demonstrated a significant specificity towards spermidine and spermine, compared with GlnA2 (Figure 4).

Assuming the Michaelis–Menten model, the Km values of GlnA2 were determined to be 0.55 ± 0.1 mM for putrescine, 1.06 ± 0.2 mM for glutamate, and 0.13 ± 0.1 mM for ATP. The Vmax value of 0.25 ± 0.05 nmol Pi per min per 1 μg of enzyme was determined for glutamate, of 0.51 ± 0.05 nmol Pi for putrescine, and 0.33 ± 0.05 nmol Pi per min per 1 μg of GlnA2 for ATP (Appendix A). Furthermore, incubation of GlnA2 with the reaction mixture containing a potent GS inhibitor methionine sulfoximine (MSO) did not affect the activity of GlnA2 (Appendix A).

### 2.3. Expression of glnA2 Is Induced by Polyamines

Previous studies in *S. coelicolor* reported that under defined glutamate depletion conditions the transcription levels of the *glnA2* and *glnA3* genes were similar, remained constant, and were lower than that of *glnA* [32]. To test whether the expression pattern of *glnA2* might be influenced by polyamines, a transcriptional analysis (RT-PCR) of *glnA2* was performed under various N-conditions. Total RNA was isolated from *S. coelicolor* M145 grown in defined Evans medium supplemented either with polyamines (25 mM), ammonium chloride (25 mM), or glutamine (5–50 mM) as a sole nitrogen source. Reverse transcription/PCR analysis was performed using primers internal to *glnA2*. The *hrdB* gene that encodes the main sigma factor of RNA polymerase and has constant levels of expression throughout growth [33] was used as internal control.

Transcriptional analysis of *glnA2* in the presence of polyamines revealed increased expression levels of the *glnA2* gene in medium with putrescine, cadaverine, and spermidine as the only N-source (Figure 5B). Such enhanced expression was previously also observed for *glnA3* in the presence of polyamines [1]. These results indicate that putative substrates of the GlnA2 enzyme induce the expression of *glnA2*.

In order to confirm the absence of the influence of glutamine on the *glnA2* expression, a luciferase-based reporter gene analysis under different nitrogen conditions was performed. To do so, the luciferase encoding genes *luxAB* lacking a promoter were fused to the approximately 200 bp long promoter region of *glnA2* and integrated into the genome of *S. coelicolor* M145. The luminescence was measured in samples incubated in defined media with sole nitrogen sources ammonium, nitrate, glutamate, and glutamine. The assay demonstrated no expression of *glnA2* in the presence of 5 mM or 50 mM glutamine as the only nitrogen source (Appendix A).

In contrast, enhanced expression of *glnA2* was observed under starvation conditions with low glucose and ammonium concentration (Figure 5C), but no expression was detected in the medium with high amounts of ammonium or glutamine (Figure 5A,C). This expression pattern is similar to that reported for *glnA3* [1], *glnA4* [25], *glnA,* and *glnII* [34] under these conditions.

### 2.4. The Deletion of glnA2 in *Streptomyces coelicolor* Resulted in Growth Defects under High Polyamine Concentrations

In order to elucidate the role of GlnA2 in the physiology of the cell, a Δ*glnA2* deletion mutant was generated. The *glnA2* gene was interrupted by the apramycin resistance gene, which was introduced into the genome by double crossover using the flanking regions of the *glnA2* gene. The deletion of *glnA2* was confirmed by the successful genetic complementation of the mutant with the wild-type *glnA2* gene. For subsequent phenotypical analysis, the *glnA2* mutant was tested for its ability to utilize different nitrogen sources. The Δ*glnA3* [21] and Δ*glnA4* [1,25] mutants as well as the parental strain *S. coelicolor* M145 were also included in this analysis. In the natural soil habitat of *S. coelicolor*, concentrations of polyamines at sites of decomposition of organic material coupled with elevated levels of these compounds are unknown. Thus, we tested the growth of mutant strains in the presence of very high concentrations of polyamines (up to 100 mM), in order to force the cells to take up polyamines as an N-source aiming to observe clear growth defects. The actual intracellular polyamine concentration was afterwards checked by HPLC analysis (see Section 2.5).

First, to investigate the potential toxicity of high polyamine concentrations on *S. coelicolor*, the growth of the mutants and the parental strain was tested on complex LB medium supplemented with polyamines at different concentrations. Growth and morphology of the parental strain and the Δ*glnA2*, Δ*glnA3*, and Δ*glnA4* mutants were monitored on LB-agar after 5 days of incubation at 30 °C. Growth of the *glnA2* and *glnA3* mutant, but not that of the *glnA4* mutant, was inhibited in the presence of a 100 mM mixture of polyamines putrescine, cadaverine, spermidine, and spermine (25 mM each) (Figure 6A), leading to complete absence of growth in the joint presence of all polyamines. Interestingly, if only putrescine or cadaverine was present in the medium, all mutants could survive, although the *glnA2* mutant showed a strongly delayed growth with cadaverine. In the presence of high concentrations of spermidine or spermine (100 mM), the growth of the *glnA2* as well as of the *glnA3* mutant was inhibited (Figure 6A). These results indicate that the disruption of *glnA2*, and not only that of *glnA3*, abolished the survival of *S*. *coelicolor* in the presence of high polyamine concentrations in the complex LB medium. The growth defect of the *glnA2* mutant observed in this study revealed that this gene might encode a functional homolog of GlnA3 required particularly for survival under high polyamine concentrations.

In order to investigate the role of *glnA2* utilizing polyamines as a C- and N-source under starvation conditions, the phenotype of the *glnA2* mutant was investigated on defined Evans-agar supplemented with the following nitrogen sources as sole N-source: ammonium (100 mM), polyamines putrescine (100 mM), cadaverine (100 mM), spermidine (100 mM), spermine (50 mM), glutamine (50 mM), nitrate (100 mM), glutamate (50 mM), and urea (50 mM). Growth of the parental strain and the mutants was monitored on defined Evans-agar after 5 days of incubation at 30 °C. The *glnA2*, *glnA3*, and *glnA4* mutants could grow with glutamate, glutamine, ammonium, nitrate, and urea as sole N-source (Figure 6B). The parental strain *S*. *coelicolor* M145 as well as the *glnA4* mutant could grow on all tested nitrogen sources (Figure 6B,C). The *glnA2* mutant demonstrated reduced growth and no sporulation in the presence of the polyamines putrescine, spermidine, and spermine and no growth with cadaverine as a sole N-source. The *glnA3* mutant revealed no growth in the presence of the polyamines putrescine (200 mM), cadaverine (100 mM), spermidine (100 mM), and spermine (50 mM). However, under 100 mM putrescine concentration, the *glnA2* and *glnA3* mutants could grow equally, though demonstrating delayed growth compared with the parental strain (Figure 6C). The complementation of the *glnA2* and *glnA3* mutants with a single gene copy under control of the strong constitutive promoter *PermE* restored the growth of these mutants on polyamine plates (Appendix A).

All phenotypic analyses of the *glnA2* mutant revealed strongly delayed or no growth with polyamines as the only N-source. The results indicate that GlnA2 is involved in polyamine metabolism. Considering growth features of investigated mutant strains we could conclude that spermine and spermidine are better substrates for GlnA3 than for GlnA2, whereas putrescine is a better substrate for GlnA2 than for GlnA3. At the same time cadaverine seems to be a bad substrate for both GlnA2 and GlnA3. These data are consistent with and thus support those from in vitro assays (see Section 2.2).

### 2.5. The glna2 Deletion Mutant Accumulates the Polyamine Cadaverine Intracellularly

In order to obtain further information on the role of GlnA2 in polyamine utilization as well as the origin of polyamine toxicity, the intracellular polyamine content of the parental strain *S. coelicolor* M145, the *glnA2* mutant, and additionally of the *glnA3* mutant and the *glnA4* mutant was analyzed by RP-HPLC. Due to the key role of the gamma-glutamylpolyamine synthetases GlnA2 and GlnA3, their deletion would block the first step of the polyamine utilization pathway leading to the change in the intracellular physiological concentrations of polyamines followed by the excess and toxicity for cells of these compounds. The strains were cultivated in a pre-culture with the complex S-medium for 4 days at 30 °C. Cells were harvested and washed with Evans medium without N-sources. The washed cells were then transferred into defined Evans medium containing the polyamines putrescine, cadaverine, or spermidine as a sole N-source and incubated at 30 °C for further 4 days. Samples for biomass determination and extraction of intracellular polyamines were collected every 24 h. Cell pellets were used to extract intracellular polyamines.

Comparison of intracellular polyamine levels during 4 days of cultivation revealed that the intracellular putrescine and spermidine concentrations in the wild type are higher than in the *glnA2* mutant. The concentrations were decreasing over the time in the parental strain and in the *glnA2* mutant, in which putrescine and spermidine disappeared almost completely. Surprisingly, putrescine disappeared rapidly in the *glnA2* mutant, pointing out the importance of GlnA3 as an enzyme indispensable for polyamine glutamylation in *S. coelicolor*. This observation is in agreement with the previous study [1] and our observations (Appendix A). However, 5-fold higher levels of cadaverine were observed in the *glnA2* mutant compared with the parental strain after 4 days of cultivation (Figure 7). The cadaverine concentration in the parental strain was decreasing over four days to one third; that of the *glnA2* mutant showed an increase of almost 70% (Figure 7). The *glnA3* deletion was demonstrated to result in the intracellular accumulation of putrescine, cadaverine, and spermidine [1] (Appendix A); the *glnA2* deletion resulted in the intracellular accumulation of cadaverine only. The *glnA4* deletion lead to no change in the polyamine utilization compared with the wild type (Appendix A), which was also proved by previous studies showing that GlnA4 is not involved in polyamine metabolism [25].

### 2.6. GlnA2 and GlnA3 Are Controlled by Different Regulatory Mechanisms

Previous studies on the regulation of nitrogen metabolism in *S. coelicolor* revealed that the global transcription regulator GlnR binds to the promoter regions of various genes involved in nitrogen metabolism, such as *glnA*, *amtB*, *glnK*, and *glnD* [35]. However, it has been demonstrated by gel retardation analysis that GlnR does not bind to the promoter regions of *glnA2*, *glnA3*, and *glnA4* [35]. Subsequent studies revealed that *glnA4* is regulated by the transcriptional regulator EpuRI [25], while the regulation of *glnA2* and *glnA3* remained unknown.

*In silico* analysis of the promoter region of *glnA2* surprisingly revealed the presence of the motif gTnAc, which previously was described to be involved in GlnR binding. In order to investigate whether conditions exist under which GlnR may bind the *glnA2* promoter region, electrophoretic mobility shift assays (EMSA) were carried out. The band shift assays performed with amplified, Cy5-labeled, and purified promoter regions of *glnA2* and Strep-GlnR delivered a visible band shift for the *glnA2* promoter region after addition of higher amounts of Strep-GlnR that was not the case when low amounts of the protein were added (Figure 8A). Since GlnR can be acetylated in the presence of nitrate as a sole nitrogen source resulting in changes of the binding affinity of GlnR [34], we aimed to also investigate the binding of acetylated GlnR to the *glnA2* promoter. Therefore, Strep-GlnR was heterologously produced in a strain that was cultured for 36 h in the defined Evans medium containing 5 mM NaNO_3_ as an N-source. The following lysine residues were acetylated in Strep-GlnR in the Evans medium: Lys 142, Lys 153, Lys 159, and Lys 200 [34]. Band shift assays revealed a strong interaction between acetylated GlnR and the *glnA2* promoter region (Figure 8A), indicating that the acetylated version of GlnR binds better to the *glnA2* promoter region than the non-acetylated version of GlnR.

In order to investigate, whether the *glnA3* gene can also be regulated by GlnR, EMSA assays were performed with GlnR and the *glnA3* promoter region. Under conditions equivalent to the experiment with the *glnA2* promoter, no band shift occurred (data not shown).

To obtain a deeper insight in the different regulatory processes of *glnA2* and *glnA3*, we searched for putative regulator genes of *glnA3*, which were assumed to be located in close proximity of *glnA*-like genes. Thereby *sco1614* and *sco5656* genes that encode regulatory proteins EpuRI and EpuRII, respectively, were identified. His-EpuRI and His-EpuRII were first heterologously produced in *E. coli* BL21 and afterwards purified by affinity chromatography. The interaction of these proteins was also tested with promoters of other genes of the polyamine utilization pathway: *sco5676*, *sco5977*, *sco6960*. These potential targets were identified based on their localization in the genome closely to genes that are involved in the utilization of polyamines as reported by [1].

The tests with EpuRII resulted in eight positive hits for EpuRII-interacting promoter sequences (Figure 8B). In addition to *glnA3*, the following genes seem to be regulated by EpuRII: *sco5676*, coding for a putative homologue of the 4-amino-butyrate aminotransferase GabT, *sco5977* encoding a putative polyamine antiporter, and *sco6960* with unknown function (Figure 8B). The interacting promoter regions also involved those of all four regulators *epuRI*, *epuRII*, *epuRIII*, and *epuRIV*, hinting towards a complex regulatory network as well as self-regulation of EpuRII (Figure 8B).

EMSA analysis with EpuRI revealed eight positive interactions with promoter regions, which include the genes encoding for the gamma-glutamylethanolamide synthetase GlnA4 as well as the regulators EpuRII and EpuRIV (Appendix A). In addition, the gene *sco5652*, which has an unknown function and the adjacent operon including *sco5654*, a putative ABC transporter, were among the positive hits. Furthermore, the promoter region of *sco5657*, encoding a putative aldehyde dehydrogenase, appeared as a target of EpuRI. Lastly the promoter sequence of epuRI itself was among the targets, indicating a self-regulating mechanism (Appendix A).

The analysis of potential regulatory targets of EpuRI and EpuRII by EMSA hint towards a complex regulatory network controlling several mono- and polyamine-associated genes, the functions of which range from utilization (*glnA3*, *glnA4*, *sco5657*, and *sco5676*) over uptake of polyamines or polyamine-containing products (*sco5654* and *sco5977*) to yet unknown purposes (*sco5652* and *sco6960*). Additionally, targets that include the regulator genes *epuRI*, *epuRII*, and *epuRIV* were found to interact with both regulators EpuRI and EpuRII. Although GlnA2 and GlnA3 are both involved in polyamine metabolism, the transcriptional regulation of their encoding genes is complex and controlled by different regulators.

### 2.7. Streptomyces tsukubaensis Can Survive Polyamine Excess Using GlnA2

The majority of *Streptomyces* sp. possess three *glnA*-like genes. However, there are strains that lack one of these genes. For example, *S. tsukubaensis* can metabolize polyamines, but a homolog of the *S. coelicolor* gamma-glutamylpolyamine synthetase GlnA3 (SCO6962) is absent. Possible “*glnA3*-candidate” proteins have less than 26% amino acid sequence identity compared with SCO6962. *S. tsukubaensis* has only two orthologs of the GS-like proteins: GlnA2 (B7R87_07810), 94% amino acid sequence identity with SCO2241, and GlnA4 (B7R87_04800), 85% amino acid sequence identity with SCO1613; whereby, GlnA4 (B7R87_04800) is a homolog of the gamma-glutamylethanolamid synthetase from *S. coelicolor*.

To analyze, whether *S. tsukubaensis* can metabolize polyamines though lacking GlnA3, the growth of the *S. tsukubaensis* NRRL18488 wild type was tested in media containing putrescine, spermidine, or spermine. The polyamines (25 mM each) were added to the defined MG medium optimized for the production of FK-506, a secondary metabolite produced naturally by *S. tsukubaensis*. Although the strain was still able to produce FK-506, the growth of *S. tsukubaensis* appeared to be significantly reduced in the presence of all polyamines (Appendix A). However, despite the lack of the *glnA3* gene in *S. tsukubaensis*, the strain could still survive in the presence of high polyamine concentrations, pointing out the ability of GlnA2 alone to utilize polyamines.

Interestingly, the growth of *S. tsukubaensis* as well as the FK-506 production were affected differently depending on the polyamine present in the medium. The presence of the polyamine putrescine resulted in both higher biomass accumulation and higher FK-506 production compared with the media supplemented with long-chain polyamines spermidine and spermine. Furthermore, very poor growth of *S. tsukubaensis* was observed in the media supplemented with 25 mM spermidine and spermine (Appendix A), while another soil *Streptomyces* strain *S. coelicolor* was able to tolerate up to 50 mM spermine and 100 mM spermidine in complex and defined media (see Section 2.4). These results indicate that *Streptomyces sp.* containing GlnA3 seem to have an evolutionary advantage in contrast with strains that lack this enzyme.

## 3. Discussion

### 3.1. GlnA2 Is a Gamma-Glutamylpolyamine Synthetase Involved in the First Step of Polyamine Utilization

In this work we demonstrated that GlnA2 is a gamma-glutamylpolyamine synthetase in *S. coelicolor*. This enzyme is a functional homolog of GlnA3 and is involved in the first step of polyamine utilization (Figure 1). Our study confirms earlier reports about the absence of the GS activity of GlnA2 in *S. coelicolor* [1,21], *C. glutamicum* [36], and *M. tuberculosis* [37]. Furthermore, the enzymatic function of GlnA2 demonstrated in this work is similar to the function of GlnA2 in the halophilic archaea *Haloferax mediterranei*. It has been demonstrated that in this organism GlnA2 is not a GS but a glutamate-putrescine ligase involved in polyamine metabolism [28].

Other reports describe the role of GlnA2 in *Halobacillus halophilus*. In this halophilic bacterium, the transcription of *glnA2* is influenced by NaCl and GlnA2 can act as a GS participating in the synthesis of glutamine as compatible solute [38]. In contrast, in *S. coelicolor* salt concentrations up to 1 M NaCl had no influence on *glnA2* transcription.

Interestingly, the phenotypical analysis of the individual *glnA2* and *glnA3* mutant in the presence of 100 mM putrescine resulted in the growth of each strain. Previous studies reported that the *glnA3* mutant was not able to grow on 200 mM putrescine in contrast with the wild type and the *glnA2* mutant [1]. These results indicate that GlnA3 is indispensable in *S*. *coelicolor* cells for polyamine utilization at high concentrations. On the other hand, our in vitro and in vivo experiments strongly suggest that spermine and spermidine are better substrates for GlnA3 than for GlnA2, whereas putrescine is a better substrate for GlnA2 than for GlnA3 and cadaverine is a bad substrate for both GlnA2 and GlnA3.

In this study we demonstrated that GlnA2 is conducting a glutamylation reaction by addition of glutamic acid to polyamines. The glutamylation reaction is common in nature and can be catalyzed by some other types of enzymes depending on the substrate and acceptors, e.g., gamma-glutamyltransferases catalyze the transfer of gamma-glutamyl groups from molecules, such as glutathione, to an acceptor that may be an amino acid or a peptide [39]. Gamma-glutamyl compounds are not rare in nature including glutathione (gamma-glutamylcysteinylglycine), poly-gamma-glutamic acid, glutamylated proteins and others [40]. The ability of GlnA2 to glutamylate monoamines and amino acids that are precursors in secondary metabolite synthesis, such as phenylalanine and cysteine, was investigated in this work. Our in vitro assays revealed that GlnA2 remains rather not active in the presence of phenylalanine and other tested monoamines or amino acids (Appendix A). This result indicates that it is unlikely that GlnA2 might be directly involved in the secondary metabolite synthesis. However, GlnA2 demonstrated some level of activity in the presence of cysteine. This might be considered for further biotechnological applications, since gamma-glutamylation of cysteine was reported to strongly increase its solubility in water [41], which is critical for the use of the compound in medication.

### 3.2. S. coelicolor Possess a Complex Network of Polyamine Utilization Pathways

Our study extends and complements the knowledge about catalytic steps in the polyamine catabolism in *S. coelicolor*. The essential steps of polyamine utilization pathways are similar to those known in *E*. *coli* and *P. aeruginosa* (Figure 1). The first glutamylation step in the gamma-glutamylation pathway in *S*. *coelicolor* requires GlnA3 and GlnA2 (Figure 1). In this study, different substrate specificity of GlnA2 and GlnA3 towards different polyamines was observed. Results obtained in this work suggest that utilization of high polyamine concentrations require both gamma-glutamylpolyamine synthetases GlnA2 and GlnA3, whereas GlnA3 is indispensable for polyamine metabolism in *S. coelicolor*. The presence of multiple glutamylation enzymes PauA1–PauA7 specific for different mono- and polyamines has also been previously demonstrated in *P. aeruginosa* [12,42,43].

In the next utilization step, polyamines might be converted to gamma-aminobutyraldehydes (Figure 1). The production of intermediate products resulting from polyamine catabolism, such as aminobutyraldehyde, can be toxic for the cell [19]. Our results in combination with a previous report [1] suggest that the conversion of these intermediates to gamma-aminobutyric acid (GABA) at different stages of polyamine utilization in *S. coelicolor* (Figure 1) is similar to those of *P. aeruginosa* and *E. coli*. The synthesis of semialdehydes might involve the aminotransferase SCO5676. It is a predicted homolog of the GABA aminotransferase GabT from *E. coli*, which catalyzes the production of glutarate semialdehyde or succinate semialdehyde. It has been shown that the expression of the *sco5676* gene is strong in the presence of arginine [44], which is a precursor of putrescine biosynthesis in *S. coelicolor*. The enhanced expression of *sco5676* in the presence of polyamines [1], as well as the interaction of the regulator of polyamine genes EpuRI with the promoter region of SCO5676 reported in this work, suggest a role of SCO5676 in the conversion of GABA to semialdehydes (Figure 1). The RNAseq results obtained in this work indicate that a predicted homolog of the glutarate/succinate semialdehyde dehydrogenase encoded by *sco5679* might catalyze the last step of putrescine, cadaverine, and spermidine glutamylation (Appendix A, Figure 1).

In *S. coelicolor*, alternative pathways may be involved in the utilization of putrescine, spermidine, and spermine, but not for the utilization of cadaverine. The utilization of putrescine can occur not only via the gamma-glutamylation pathway, but also via the aminotransferase pathway or direct oxidation pathway, as in *P. aeruginosa*. A putative amidotransferase (SCO5655) was identified in silico and is predicted to be homolog of the putrescine amidotransferase (PatA) from *E. coli*. In transcriptional analysis, the expression of *sco5655* was enhanced in the presence of polyamines [1]. Moreover, *sco5655* was reported to be induced by a diamide [45] and not by arginine [44]. This indicate that in *S. coelicolor*, putrescine may be catabolized also via the alternative aminotransferase pathway. Additionally, spermidine and spermine may be further reduced by the predicted gamma-glutamylpolyamine oxidoreductase (SCO5671), which is a close ortholog of the gamma-glutamylpolyamine oxidoreductases PuuB from *E. coli* and PauB1–B4 from *P. aeruginosa*. Interestingly, the gene *sco5671* was expressed in the presence of spermidine but not in the presence of the polyamines putrescine and cadaverine [1]. SCO5671 might convert spermidine and spermine to gamma-aminobutyraldehyde in the spermine/spermidine dehydrogenase pathway. Besides that, spermidine might also be acetylated by a predicted acetyltransferase homolog SCO3913 (Figure 1).

In summary, *Streptomyces* apparently possess a complex network of enzymes and regulators for metabolizing the different polyamines at different concentrations. For *Streptomyces*, this network can be summarized by a model including all known and suggested enzymes of polyamines metabolism (Figure 1). The complexity of the network is also reflected by the occurrence of different metabolic pathways and numerous regulators contributing to the transcriptional control of the various metabolic genes. However, a central role seemed to be played by the two gamma-glutamylpolyamine synthetases GlnA2 and GlnA3 that initiate polyamines metabolization.

## 4. Materials and Methods

### 4.1. Strains and Growth Conditions

*S. coelicolor* M145 (parental strain) and the mutants were incubated for 5 days at 30 °C in rich LB-agar or in defined Evans-agar [46] supplemented with different nitrogen sources: monosodium glutamate, l-glutamine, ammonium chloride, sodium nitrate, urea, cadaverine dihydrochloride, putrescine dihydrochloride, spermidine trihydrochloride, and spermine tetrahydrochloride in a concentration range of 5–200 mM. Growth experiments in liquid culture were performed using either complex S-medium [47] or chemically defined Evans medium (modified after [46]). Media were supplemented with polyamines or ammonium chloride as a sole nitrogen source. Strains were cultivated at 30 °C on the rotary shaker (180 rpm) for 5 days. Genetic manipulation of *S. coelicolor* M145 was carried out as described by [48,49]. For preparation of genomic DNA, *S. coelicolor* M145 was cultivated for 4 days in S-medium and DNA was isolated with the NucleoSpin Tissue Kit (Macherey-Nagel, Düren, Germany).

*S. tsukubaensis* NRRL 18488 was obtained from the NRRL Culture Collection of the Agricultural Research Service (1815 N. University Street Peoria, IL, USA). Spores and mycelia preparations were obtained from ISP4 (Difco, Sparks, MD, USA) agar plates. FK-506 production by *S. tsukubaensis* NRRL 18488 was analyzed in the liquid media MG optimized by [50].

All strains and plasmids used in this study are listed in the Table 2.

### 4.2. Construction of the ∆glnA2 Deletion Mutant

For the generation of the ∆*glnA2* deletion mutant, a 2 kb DNA region was amplified by PCR from *S. coelicolor* genomic DNA covering 1.15 kb of the *glnA2* upstream region and 850 bp of the *glnA2* gene. The PCR product was integrated into the pDrive plasmid and sequenced. Then it was cloned into the plasmid pK18, which does not replicate in Streptomyces, via the restriction sites *Eco*RI and *Hin*dIII. The apramycin resistance cassette was amplified from the pUC21 plasmid and inserted into the DNA fragment for the *glnA2* gene, disrupting the coding sequence after the first 150 bp. The resulting plasmid pK18/*glnA2*-Apra was used for transformation of *S. coelicolor* M145 using protoplasts. Resistant colonies were plated on LB agar with apramycin or kanamycin to select the clones in which the integration of the DNA fragment into *glnA2* by means of double crossover had taken place. Single crossover clones containing the entire plasmid pK18/*glnA2*-Apra were both apramycin and kanamycin resistant, whereas double crossover clones possessed only the apramycin resistance. *glnA2* mutants were verified by PCR and sequencing.

All primers used in this study are listed in the Appendix A.

### 4.3. Construction of ∆glnA2 Complementation Strains

For a complementation experiment, the *glnA2* gene without its native promoter was cloned in the multiple cloning site of the integrative plasmid pRM4 [52] behind the constitutively expressed *_P_ermE* promoter. Afterwards, the plasmid was introduced into the ∆*glnA2* mutant by biparental conjugation using the *E. coli* S17 strain. Clones were then selected by resistance against kanamycin and apramycin, then confirmed by PCR and sequencing.

### 4.4. Cloning, Production, and Purification of His-GlnA2, His-EpuRI, and His-EpuRII

The GlnA2 encoding gene *sco2241*, the EpuRI encoding gene *sco1614,* and EpuRII encoding gene *sco5656* were amplified by PCR from *S. coelicolor* genomic DNA and ligated into the expression vector pJOE2775 [54] under the control of rhamnose inducible promoter *_P_rha*. The restriction sites *Nde*I and *Hin*dIII were used for cloning. His-GlnA2, His-EpuRI, and His-EpuRII were produced in the *E. coli* strain BL21 (DE3). Cells were incubated in a pre-culture containing the rich LB medium at 37 °C until the culture density reached an optical density of 0.6 at 600 nm. The bacterial cells were then induced by 0.2% rhamnose and incubated overnight. Afterwards, the cells were harvested by centrifugation and stored at −20 °C until needed. His-GlnA2, His-EpuRI, and His-EpuRII were purified by nickel ion affinity chromatography as described by the resin manufacturer (GE-Healthcare GmbH, Solingen, Germany). Purified His-GlnA2 was dialyzed against 20 mM Tris and 100 mM NaCl (pH 8) and immediately used for further analysis.

### 4.5. Cloning, Production, and Purification of Strep-GlnR

GlnR was used for promoter binding studies with an N-terminal StrepII-Tag. *glnR* was amplified with primers that add a sequence encoding an N-terminal StrepII-tag using Taq polymerase (Qiagen, Hilden, Germany), and subcloned into the pDrive cloning vector (Qiagen, Hilden, Germany). After the digestion with *Nde*I and *Hin*dIII, Strep-*glnR* was cloned into pJOE2775 under the control of the *_P_rham* promoter, resulting in the pYT9 plasmid. Gene expression was induced with 0.2% rhamnose for 12 h. Strep-GlnR was produced in *E. coli* Xl1blue strain [51] and afterwards purified using StrepTactin Superflow gravity-flow columns (IBA, Göttingen, Germany) [53].

### 4.6. In Silico Protein Modeling

To build the in silico models for the different GlnA-like enzymes an existing template from the Protein Data Bank (PDB) was selected. The initial template search was carried out using the template search function of SWISS-MODEL [56] by input of the amino acid sequence in FASTA format, plain text, or UniProtKB accession code. SWISS-MODEL then performed a search for evolutionary related protein structures against the SWISS-MODEL template library SMTL [57] and used database search methods BLAST and HHblits. The resulting template structures were ranked and further evaluated by SWISS-MODEL using the estimated global model quality estimate (GMQE) [57] and the quaternary structure quality estimate (QSQE) [58]. Top-ranked templates and alignments were compared to verify whether they represented alternative conformational states or covered different regions of the target protein. In such case, multiple templates were selected automatically and different models were built accordingly [56]. Out of the resulting list of possible templates, different templates were chosen based on the GMQE, QSQE, identity, and oligo state. In addition, value was given to select templates out of different bacterial phyla to obtain quality control with diversity. Using the selected templates as a base, SWISS-MODEL built a 3D protein model estimating the real 3D structure of the protein. Therefore, SWISS-MODEL started with the conserved atom coordinates defined by the target-template alignment and then coordinated residues corresponding to insertions/deletions in the alignment that were generated by loop modelling, and a full-atom protein model was obtained by constructing the non-conserved amino acid side chains [56]. SWISS-MODEL used the ProMod3 modelling engine and the OpenStructure computational structural biology framework [57]. The evaluation of the build 3D protein models was conducted using the QMEAN scoring function, the MolProbility score, and a Ramachandran plot of the model. The QMEAN score provided an estimate of the “degree of nativeness” of the structural features observed in a model and described the likelihood that a given model was of comparable quality compared with experimental structures [59]. The MolProbility score relied heavily on the power and sensitivity provided by optimized hydrogen placement and all-atom contact analysis, complemented by updated versions of covalent-geometry and torsion-angle criteria [60]. The Ramachandran plot plotted the torsion angles of the different amino acids against each other to verify the correct folding of the in silico model [61]. Based on the highest QMEAN score and the Ramachandran plot, the best in silico 3D protein models were chosen.

### 4.7. HPLC/ESI-MS Detection of the Glutamylated-Product

For the detection of the glutamylated product of the GlnA2 catalyzed reaction, a HPLC/ESI-MS procedure was applied. Standard reactions contained: 20 mM HEPES (pH 7.2), 10 mM ATP, 150 mM glutamate sodium monohydrate, 150 mM putrescine dihydrochloride, cadaverine dihydrochloride, spermidine trihydrochloride, or spermine tetrahydrochloride, 20 mM MgCl_2_ × 6H_2_O—mixed with 10 µg of the purified His-GlnA2 (or without GlnA2 as a control) and incubated at 30 °C for 5 min. The reaction mixture was incubated at 100 °C for 5 min in order to stop the reaction.

HPLC/ESI-MS analysis was performed on an Agilent 1200 HPLC series using a Reprosil 120 C_18_ AQ column, 5 µm, 200 mm by 2 mm fitted with a precolumn 10 mm by 2 mm (Dr. Maisch GmbH, Ammerbuch, Germany) coupled to an Agilent LC/MSD Ultra Trap System XCT 6330 (Agilent, Waldbronn, Germany). For analysis were used: 0.1% formic acid as solvent A and acetonitrile with 0.06% formic acid as solvent B at a flow rate of 0.4 mL min^−1^. The gradient was as follows: t_0_ = t_5_ = 0% B, t_20_ = 40% B (time in minutes). Injection volume was 2.5 µL; column temperature was 40 °C. ESI ionization was carried out in positive mode with a capillary voltage of 3.5 kV and a drying gas temperature of 350 °C.

### 4.8. Determination of the Intracellular Polyamine Content by HPLC

Amounts of the intracellular polyamines were analyzed by high-performance liquid chromatography (HPLC), as described by [62]. The method was modified by adding a step for cell disruption using glass beads and optimized for polyamine extraction and analytics in *S. coelicolor*. Cells were harvested by centrifugation (6000× *g*, 10 min at 4 °C) and washed three times in phosphate-buffered saline (PBS). An amount of 0.5 g of the wet cells was resuspended in morpholinepropanesulfonic acid (MOPS) lysis buffer (100 mM MOPS, 50 mM NaCl, 20 mM MgCl_2_, pH 8.0) and disrupted using glass beads (150–212 µm, Sigma) in a Precellys homogenizer (6500 r.p.m., 20–30 s; Peqlab). Afterwards, samples were centrifuged (13,000× *g*, 10 min at 4 °C), and trichloroacetic acid was added to the supernatant to a final concentration of 10%. The mixture was incubated on ice for 5 min and then cleared by centrifugation (13,000× *g*, 10 min at 4 °C). The pH of each sample was neutralized using HCl, and the samples were stored at −20 °C until analysis. Polyamines were derivatized using pre-column derivatization with ortho-phthalaldehyde (OPA)/mercaptoethanol (MCE) (Dr. Maisch GmbH, Ammerbuch). OPA-derivatized polyamines were separated on a Reprosil OPA column (150 × 4.0 mm, 3 µm) fitted with a precolumn (10 × 4 mm, same stationary phase) (Dr. Maisch GmbH, Ammerbuch) on a HP1090 liquid chromatograph with a diode-array detector, a thermostated autosampler, and a HP Kayak XM 600 ChemStation (Agilent, Waldbronn). UV detection was performed at 340 nm. The flow rate of 1.1 mL/min was applied with a gradient as follows: solvent A (25 mM sodium phosphate buffer, pH 7.2, containing 0.75% tetrahydrofuran) and solvent B (25 mM sodium phosphate buffer, pH 7.2 (50%), methanol (35%), acetonitrile (15%) by volume): t_0_ = 60% B, t_2.__5_ = 70%, t_12_ = t_22_ = 100% B (time in minutes). Cadaverine dihydrochloride, putrescine dihydrochloride, spermidine trihydrochloride, and spermine tetrahydrochloride purchased from Sigma were prepared with distillated water and were used as standards.

### 4.9. Analysis of Growth and FK-506 Production

The productivity of an *S. tsukubaensis* strain was determined as production of FK-506 per g cell dry weight. During the fermentation time of 6 days, all 24 h 1 mL of bacterial culture was gathered and centrifuged. It was washed twice with H_2_O_deion_ and dried by lyophillization. For FK-506 determination, the broth samples were mixed with equal volume of ethylacetate (1:1), stirred for 10 min, and centrifuged. The organic phase was analyzed via an HP1090 M HPLC system equipped with a diode array detector as well as a thermostatic autosampler. A Zorbax Bonus RP column, 3 × 150, 5 μm (Agilent Technologies, Santa Clara, CA, USA), constituted the stationary phase. The mobile phase system was applied with 0.1% phosphoric acid/0.2% triethylamine as eluent A and acetonitrile with 1% tetrahydrofurane as eluent B. The flow rate was 850 μL/min; the column temperature was set to 60 °C. The absorbance was monitored at a wavelength of 210 nm. Data sets were processed and analyzed with the help of the Chemstation LC 3D, Rev. A.08.03 software from Agilent Technologies. Standards of pure FK-506 were obtained from Antibioticos SA, León, Spain.

### 4.10. Modified Glutamine Synthetase Activity Assay

The enzymatic activity of GlnA2 was tested in a modified glutamine synthetase activity assay [30], in which ammonium substrate was replaced by polyamines. Solutions A (12% (*w*/*v*) L-ascorbic acid in 1 N HCl), B (2% (*w*/*v*) ammonium molybdate tetrahydrate in H_2_O_deion_.), C (1% (*w*/*v*) ammonium molybdate tetrahydrate in H_2_O_deion_.), and F (2% sodium citrate tribasic dihydrate and 2% acetic acid in H_2_O_deion_.), and the reaction mix were prepared containing enough protein for the release of 35–50 mM Pi produced in 5 min. After the adjustment of the pH, 95 μL was loaded into PCR-strips for each reaction. Solution D (mixture of two parts solution A and one part solution B) or solution E (mixture of two parts solution A and one part solution C) for undetermined kinetic parameters of ATP was prepared. Afterwards, the reaction was initiated by adding 5 μL substrate to the reaction mix. Additionally, blanks with H_2_O_deion_ and a phosphate standard ranging from 0 to 20 mM were included. The reaction mix was incubated at 30 °C for 5 min in a thermocycler. The wells of a 96-well plate were loaded with 150 μL of solution D (or C) for each reaction. Afterwards, 50 μL of the reaction mix was transferred to the previously prepared solution D (or C) in the 96-well plate. Solutions were mixed and incubated for 5 min at RT. At low pH, the enzymatic reaction was terminated, while 150 μL of solution F was added to stop color development. The final reaction was incubated for 15 min at RT. The absorbance was measured at 655 nm using a microplate reader. Raw absorbance readings were put into Excel (Microsoft). The least squares fit to the Michaelis–Menten equation was calculated with Prism 6 (GraphPad Software, Inc., San Diego, CA, USA).

### 4.11. Electrophoretic Mobility Shift Assay (EMSA)

An amount of 250 bp DNA fragments containing the upstream regions of target genes was amplified with the Taq-DNA polymerase and specific EMSA primers using the PCR (Appendix A). The product was cleaned by QIAquick™ PCR Purification Kit. An amount of 2 μL of the first PCR product was used as a template for the second labelling PCR reaction. Fragment labelling was performed using Cy5 labelling primers. After the Cy5 labelling, PCR products were purified with Illustra™ MicroSpin S-400 HR Columns (GE Healthcare) and stored in a black microcentrifuge cup at −20 °C. Each DNA fragment was mixed with 0.5–2 μg of purified tested regulatory proteins and incubated at RT. To verify the specificity of the protein–DNA interaction, an excess of unlabeled, specific or non-specific DNA was added to the protein–DNA incubation mixture.

### 4.12. Gene Expression Analysis by RT-PCR

For the analysis of gene transcription, the *S. coelicolor* M145 wild type and mutants were incubated in S-medium. After 4 days, cells were washed twice with Evans medium and cultivated for 24 h in Evans medium supplemented either with 25 mM ammonium chloride or 25 mM polyamine. RNA isolation was carried out with the RNeasy kit (Qiagen, Hilden, Germany). RNA preparations were performed twice with DNase (Fermentas): an on-column digestion was carried out for 30 min at 24 °C, and afterwards RNA samples were treated with DNase for 1.5 h at 37 °C. RNA concentrations and quality were proved using a NanoDrop ND-1000 spectrophotometer (Thermo Fisher Scientific, Karlsruhe, Germany). cDNA from 3 µg RNA was generated with random nonamer primers (Sigma-Aldrich Chemie GmbH, Munich, Germany), reverse transcriptase, and cofactors (Fermentas). PCR reactions were performed with primers listed in the Appendix A. The PCR conditions were as follows: 95 °C for 5 min; 35 cycles of 95 °C for 15 s, 55–60 °C for 30 s and 72 °C for 30 s; and 72 °C for 10 min. As a positive control, cDNA was amplified from the major vegetative sigma factor (*hrdB*) transcript, which is a housekeeping gene produced constitutively. To exclude DNA contamination, negative controls were carried out by using total RNA as a template for each RT-PCR reaction.

### 4.13. Luciferase Based Reporter Gene Analysis

To perform the luciferase-based reporter gene analysis under different nitrogen conditions in *S. coelicolor* M145, the pRLux86 vector was used, which is a pIJ8660 derivate in which the *egfp* gene is exchanged by the luciferase-encoding *luxAB* gene. *luxAB* originated from *Vibrio harveyi* and encodes for luciferase alpha- and beta- sub-unit [63]. The ca. 200 bp long promoter region of *glnA2* was cloned upstream of the *luxAB* gene, which lacks a promoter. The construct containing *luxAB* and the promoter region of *glnA2* was then integrated into the genome of *S. coelicolor* M145. The strain was incubated in S medium 48 h and transferred into the defined N-Evans media with 1 mM or 100 mM NH_4_Cl, 1 mM or 100 mM NaNO_3_, or 5 mM or 50 mM glutamate. Samples were then collected, transferred to the 96-well micro titer plates, and incubated for 1–6 h. Luminescence was tested in 96-well plates in Luminometer (Polar Star* Galaxy, BMG, Labtechnologies, Ortenberg, Germany) at 490 nm.

### 4.14. RNAseq Analysis

In order to isolate total RNA from *S. coelicolor* cells, bacterial cell pellets were prepared and harvested as described in the section “Strains and growth conditions”. The samples were kept at −80 °C, thawed on ice, and RNA was extracted. Polymerase chain reactions with Taq polymerase (New England Biolabs, Frankfurt, Germany) were performed to detect whether contaminating genomic DNA remained in the samples. RNA samples with genomic DNA contamination were treated with RNase-free DNase (Qiagen, Hilden, Germany). Total RNA concentrations were measured using a spectrophotometer (NanoDrop^®^, ND-1000; Thermo Fisher Scientific, Schwerte, Germany). The extracted RNA samples were either pooled (treatment with indole) or separately treated (treatment with Ind-Ala). The ribosomal RNA molecules from the isolated total RNA were removed and checked. RNA was free of detectable rRNA. Subsequently, preparation of cDNA libraries was performed. Each cDNA library was sequenced on a Sequencer system and evaluated. Statistically significant expression changes (adjusted *p*-value ≤ 0.01) with log2 fold change > 1.5 or <−1.5 were used.

## Figures and Tables

**Figure 2 ijms-23-03752-f002:**
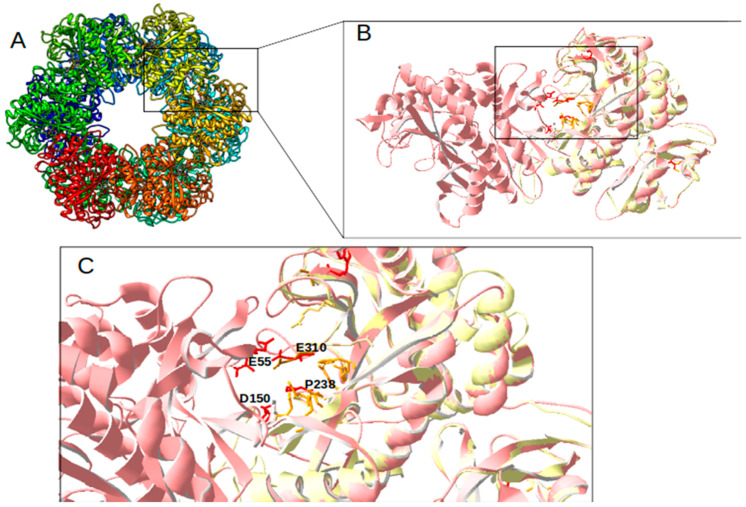
Model structure of GlnA2 from *Streptomyces coelicolor*. The overall dodecamer structure is based on the GS1 from *S. typhimurium* (PDB: 1FPY) that was used as a template (**A**). The sub-units A and B are zoomed out and depicted in light red and are superposed with the 1FPY template (yellow) (**B**). The active site is shown with the identified key amino acid substitutions in GlnA2 (red) compared with GS1 from *S. typhimurium* (dark yellow) (**C**).

**Figure 3 ijms-23-03752-f003:**
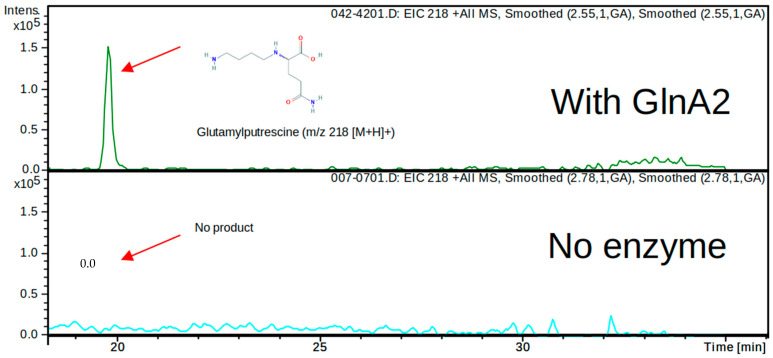
HPLC/ESI-MS analysis of the glutamylated reaction product generated by His-GlnA2. Two samples were analyzed in the MS positive mode (EIC—extracted ion chromatogram): reaction mixtures without addition of GlnA2 (below) and with addition of GlnA2 (above). Extracted ion chromatograms for the GlnA2 reaction product corresponding to gamma-glutamylputrescine with charge to mass ratio of 218 *m*/*z* is shown (above), and no product in the sample without GlnA2 could be detected (below). The chromatogram demonstrates detected peaks at the retention time 18 min and later (for full chromatogram see Appendix A).

**Figure 4 ijms-23-03752-f004:**
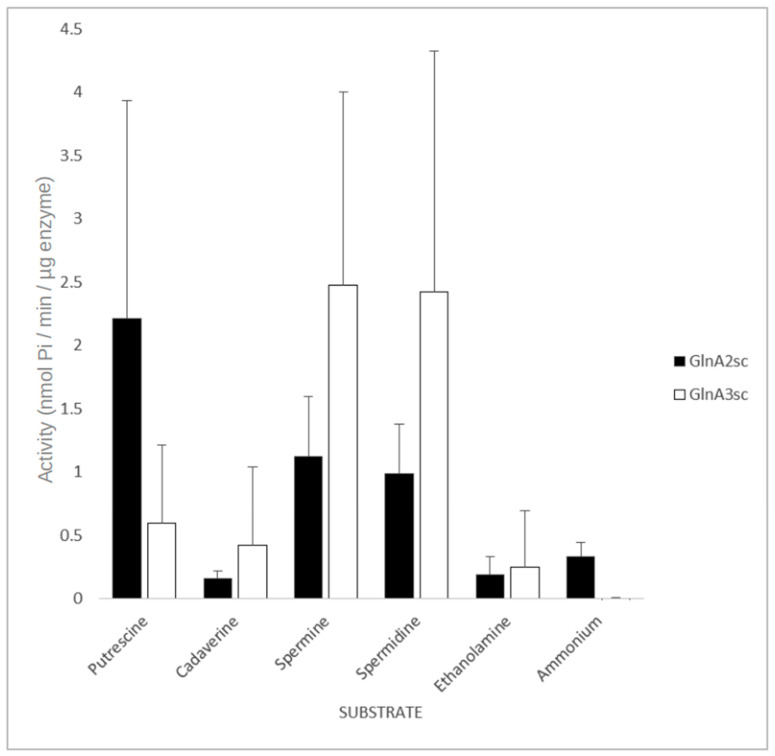
Effect of different amines on the activity of GlnA2 and GlnA3. All substrates were at a concentration of 50 mM. The mean value of *n* = 6 biological replicates from different cultures with *n* = 3 technical replicates each with standard error is shown.

**Figure 5 ijms-23-03752-f005:**
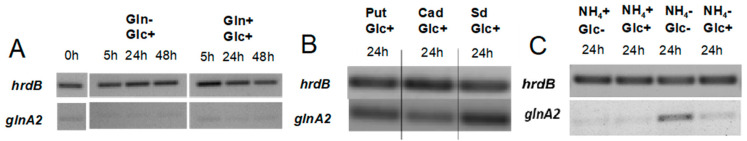
Transcription analysis of *glnA2* in *S. coelicolor* M145 under different N-conditions. RT-PCR of *glnA2* and *hrdB* in *S. coelicolor* M145 cultivated in minimal Evans medium with: (**A**) low 5 mM (Gln−) or high 50 mM (Gln+) glutamine concentration; (**B**) high 25 mM polyamine concentration (Put: putrescine, Cad: cadaverine, Sd: spermidine); (**C**) low 5 mM (NH_4_^−^) or high 50 mM (NH_4_^+^) ammonium/low 2.5 g/L (Glc−) or high 25 g/L (Glc+) glucose concentration. Total RNA was isolated from mycelium harvested after 0–48 h (**A**) or 24 h (**B**,**C**) of cultivation in the defined Evans medium.

**Figure 6 ijms-23-03752-f006:**
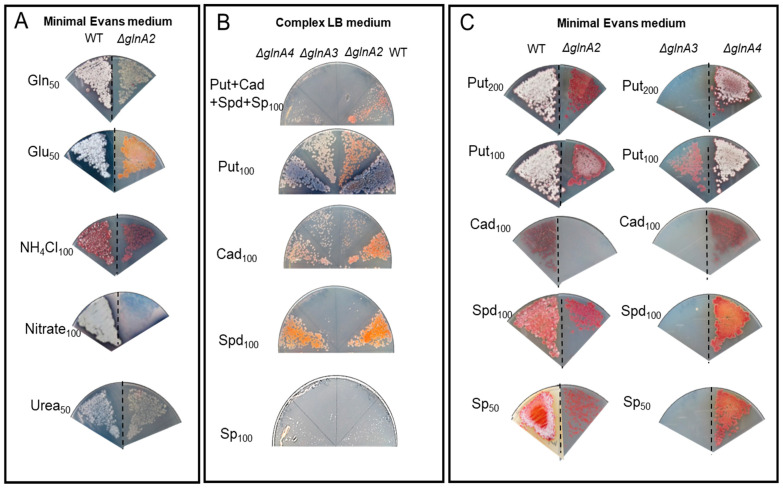
Growth of *S. coelicolor* M145 as well as *glnA2*, *glnA3*, and *glnA4* mutants on polyamine containing media. (**A**) Phenotypic analysis of strains on the rich LB-medium in the presence of 100 mM putrescine (Put), cadaverine (Cad), spermidine (Spd), and spermine (Sp) or polyamine mixture (25 mM each). (**B**) Phenotypic analysis of strains on defined Evans medium with glutamine (50 mM) (Gln), glutamate (50 mM) (Glu), ammonium (100 mM) (NH_4_Cl), nitrate (100 mM), urea (50 mM). (**C**) Phenotypic analysis of strains on defined Evans medium with putrescine (100 mM), cadaverine (100 mM), spermidine (100 mM), and spermine (50 mM). Each panel represents observations on a single agar plate; observations on separate agar plates are indicated by the dashed line.

**Figure 7 ijms-23-03752-f007:**
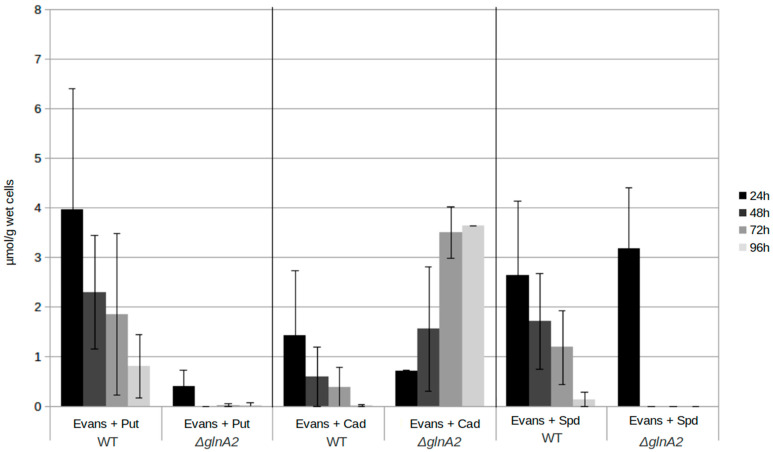
Intracellular polyamine concentration in *S. coelicolor* strains. The polyamine level of the Δ*glnA2* mutant and parental strain *S. coelicolor* M145 was monitored in samples taken after 24, 48, 72, and 96 h of cultivation in defined Evans medium supplemented with polyamines (Put—putrescine; Cad—cadaverine, or Spd—spermidine, 25 mM of each) as a sole nitrogen source. The mean value of three biological replicates was calculated in μmol per 1 g of wet cells. Error bars indicate standard error of two biological replicates.

**Figure 8 ijms-23-03752-f008:**
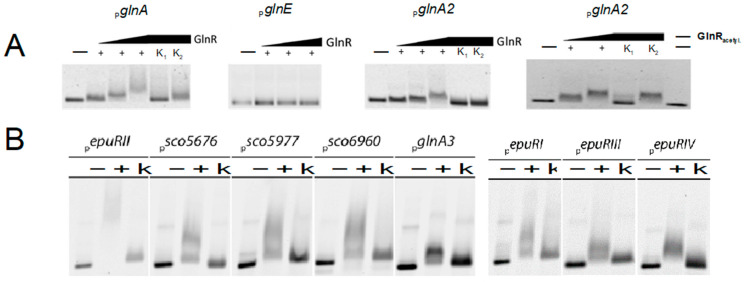
Band shift assays performed with GlnR and EpuRII. (**A**) 1 ng fluorescence-labeled *glnA2* promoter region was incubated without (−) or with (+) 0.5–2 μg Strep-GlnR (increasing arrow). As a control 1000-fold amount of unmarked specific (K1) or non-specific DNA (K2) was added. (**B**) 1 ng fluorescence-labeled promoter regions of polyamine associated genes were incubated without (−) or with (+) 2 μg His-EpuRII. As a control 1000-fold amount of specific unlabeled DNA fragments (K) was added.

**Table 1 ijms-23-03752-t001:** Comparison of amino acid substitutions in the ammonium binding site of the model structure of GlnA2 and GlnA3 from *Streptomyces coelicolor* with the GlnA from *S. typhimurium* (PDB: 1FPY).

Protein	GlnA*_St_*	GlnA2*_Sc_*	GlnA3*_sc_*
PDB	1FPY	none	none
Amino acids	Ser53	Glu55	Ser76
	Tyr179	Asp150	Ala169
	Asn264	Pro238	Asn260
	Glu327	Glu310	Thr327

**Table 2 ijms-23-03752-t002:** Strains and plasmids used in this study.

Strains/Plasmids	Genotype/Phenotype	Reference
*E. coli* XL1-Blue	*recA1, endA1 gyrA96, thi-1, hsdR17, supE44, relA1, lac* [F′, *proAB lacI^q^*ZΔ, 1M15Tn*10* Tet^R^]	[51]
*S. coelicolor* M145	*S. coelicolor* A3(2), plasmid free	[48]
*S. coelicolor* M145 ∆*glnA2*	*glnA2* mutant strain of *S. coelicolor* M145; insertional inactivation of *glnA2* by an *aac(3)IV* cassette, Apr^R^	This work
S. coelicolor M145 ∆*glnA3*	*glnA3* mutant strain of *S. coelicolor* M145; insertional inactivation of *glnA3* by an *aac(3)IV* cassette, Apr^R^	[22]
*S. coelicolor* M145 ∆*glnA4*	*glnA4* mutant strain of *S. coelicolor* M145; *glnA4* replaced by an *aac(3)IV* cassette, Apr^R^	[1,25]
*S. coelicolor* M145 ∆*glnA2*pRM4*glnA2*	Complemented *glnA2* mutant strain of *S. coelicolor* M145; Apr^R^ and Km^R^	This work
*S. coelicolor* M145 ∆*glnA3*pRM4*glnA3*	Complemented *glnA3* mutant strain of *S. coelicolor* M145; Apr^R^ and Km^R^	[1]
*S. coelicolor* M145 ∆*glnA4*pRM4*glnA4*	Complemented *glnA4* mutant strain of *S. coelicolor* M145; Apr^R^ and Km^R^	[1,25]
*Streptomyces tsukubaensis* NRRL18488	STP1 STP2	[50]
pRM4	pSET152_p_*ermE* with artificial RBS, Apr^R^	[52]
pRM4/*glnA2*	pRM4 with PCR-amplified *glnA2*, Apr^R^ and Km^R^	This work
pDRIVE	TA-cloning vector	Qiagen, Hilden, Germany
pDRIVE/*glnR*	Strep–*glnR* fragment cloned into pDRIVE	[53]
pJOE2775	pBR322-derived vector with _P_*rham* expression cassette and C-terminal His6-tag, (Amp^R^)	[54]
pIJ778	pBluescript II SK(+)-derived with *aadA* from Ω-fragment, (Spec^R^, Strep^R^) + *oriT*	[49]
pK18	pUC-derived, LacZ’ α-complementation, (Kan^R^) + *oriT*	[55]
pJOE2775/his-*epuRI*	pJOE2775-derived, over-expression of *epuRI* with 6xHis-tag, (Amp^R^)	This work
pJOE2775/his-*epuRII*	pJOE2775-derived, over-expression of *epuRII* with 6xHis-tag, (Amp^R^)	This work
pYT/his-*glnA2*	pJOE2775-derived, over-expression of *glnA2* with 6xHis-tag, (Amp^R^)	This work
pYT9/strep-*glnR*	pJOE2775 with PCR amplified Strep–*glnR* cloned *Nde*I-*Hin*dIII	[53]

## Data Availability

Data are involved in the study.

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
