# Peer review of "A Second Gamma-Glutamylpolyamine Synthetase, GlnA2, Is Involved in Polyamine Catabolism in *Streptomyces coelicolor"

_ijms, 2022, doi:10.3390/ijms23073752_

Round 1

Reviewer 1 Report

This manuscript is related to a microorganism (S.coelicolor) that can use polyamines as nitrogen sources. The authors have shown that the gene SCO2241 (glnA2) allows S.coelicolor to survive under toxic concentrations of polyamines. Also, they have shown that although this gene codifies a glutamine synthetase-like enzyme, it is a gamma-glutamylpolyamine synthetase. They have also demonstrated that glnA2 and glnA3 are involved in polyamine catabolism in S.coelicolor.

The manuscript deserves to be published and is of great interest to researchers on nitrogen metabolism since it opens the study to new metabolic routes.

The manuscript has some typos:

Line 3    Streptomyces Coelicolor   must be  Streptomyces coelicolor

Line 55  Archea must be Archaea

Line 155   thecomparison  must be the comparison

Author Response

Thank you for kind comments. Please see the attachment.

Reviewer 2 Report

Evaluation of the article “A Second Gamma-Glutamylpolyamine Synthetase, Glna2, is 2 Involved in Polyamine Catabolism in Streptomyces Coelicolor” by Krisenko et al.

 The work of Krisenko et al. is aimed at the characterization of a Gamma-Glutamyl polyamine synthetase GlnA2, of Streptomyces coelicolor that is involved in the catabolism of polyamines. This enzyme is a functional analogue of the GlnA3 enzyme described by the same authors in 2017, and the authors compare the polyamine substrate specificity of these two enzymes. The g-Glutamyl polyamine synthetase has being characterized by cloning and disrupting the encoding glnA2 gene and using chemical analysis to determine the product of the reaction but the routine measure of the enzyme activity is done by measuring the release of the Pi from ATP (see below). The authors have done also expression studies to see if the activity of the enzyme is induced by polyamines.  and describe in some detail the role of several similar enzymes of other microorganisms in the catabolism of polyamines. The subject is of interest since it helps to provides a more complete view of the complex regulation of nitrogen metabolism and its relation to polyamine catabolism. However, some points should be addressed before the article is accepted.

Specific Comments.

  1. The tittle says “Glna2”, is this correct?. The letter A should be in capital. The tittle says “Coelicolor” this is incorrectly spelled in microbiological terms. The species name is always in lower cases.
  2. In lines 57-58 the authors introduce the role of polyamines in several cellular processes including cells growth, but they fail to indicate that polyamines control the biosynthesis of secondary metabolites in several microorganisms, as occurs with the biosynthesis of penicillin. There are several references about this subject and some should be cited.
  3. In line 120 there is a sentence which as no sense, “the occurrence of GlnA2, which has 69% amino acid sequence identity to coelicolor GlnA2…”. The authors are working with S. coelicolor M145 therefore the protein should be identical.
  4. In line 213 the authors state “the enzymatic activity and substrate specificity of GlnA2, an adapted GS activity assay [28] based on the inorganic phosphate (Pi) released from ATP was used”. The release of Pi in ATP consuming reactions is relatively variable as shown by several authors; there are factors affecting the purity of the substrate and the enzyme preparation that may affect this measure. The authors have HPLC-MS facilities that would be better used in the quantification of the enzyme activity. Is not the HPLC-MS sufficiently reliable to quantify the enzyme activity?
  5. In the section 2.4 (lines 281 and following) to study the toxicity of polyamines on the glnA2-deleted mutant the authors use equal quantities (0.1M) of each of the polyamines. This is a very large polyamin concentration and I doubt that this concentration occurs in nature. Why the authors used such a large concentration of polyamines? May be better to do the studies with a moderate polyamines concentration, e.g. 10 to 50 mM, to observe the relevant toxicity effect.
  6. In lines 385-388 the they state: “Band shift assays revealed a strong interaction between acetylated GlnR and the glnA2 promoter region (Fig. 8 A), indicating that the acetylated version of GlnR binds better to the glnA2 promoter region”. How many acetyl groups have the acetylated GlnR protein? do the authors know in which amino acid residues is acetylated? This should be clarified at this point and indicated in the Materials and Methods section.
  7. In lines 435 the authors say “For example, tsukubaensis can metabolize polyamines, but a homolog of the S. coelicolor gamma-glutamylpolyamine synthetase GlnA3 (SCO6962) is absent”. An important question is whether S. tsukubaensis and other Steptomyces species that lacks this third polyamine catabolism enzyme are more sensitive to large polyamines such as spermine and spermidine. This might exclude some Streptomyces sp. from habitat containing relatively large amounts of polyamines. Please clarify this point in more detail
  8. In the Discussion the author focus on the comparative analysis of the polyamines catabolism pathway in different microorganisms. An important question is whether the g-glutamyl polyamine transferases may add glutamic acid to other amine compounds including enzymes of precursors or intermediates for secondary metabolism. I suggests that the authors consider this possibility in the Discussion.

Author Response

(The authors gave the same response as above.)

Reviewer 3 Report

REVIEW

The manuscript of Krysenko et al. entitled “A Second Gamma-Glutamylpolyamine Synthetase, Glna2, is Involved in Polyamine Catabolism in Streptomyces Coelicolor”  decipher the enzymatic  function of GlnA2 by comparison with the enzymatic activity  of GlnA3 and its 3D structure. Both enzymes are able to glutamylate various polyamines but with different efficiency. The authors also show that the expression of GlnA2 is induced in the presence of various polyamines including cadaverine that does not seem to be a good substrate for GlnA2.  The regulation of the expression of GlnA2 and GlnA3 is rather complex involving numerous regulators that the authors start to characterize using EMSA (Figure 8). They also attempt to determine the intracellular  fate of different polyamines in the WT and in GlnA2, A3 and A4  mutant strains of S. coelicolor but these data are incomplete (Figure 7) and should be completed.

The paper of Krysenko et al. involves a lot of work and is reasonably well written but some minor modifications were proposed to improve the readability of the paper. However, two main problems were identified in this manuscript. The first one (easy to correct) is that the authors did not make the most and their data (figure 4 and 6) and I think that they can go further in proposing substrate specificity for GlnA2 and GlnA3.  The second one is the incompleteness of Figure 7. This problem will probably be more difficult to correct since it would involve extra experimental work.

Other comments

- I would suggest to replace Figure 1 by figure 9 and thus to cancel actual Figure 1 since these two figures are similar. 

Section 2.2

- Can the authors explain why they did not choose the strategy used to demonstrate the ability of GlnA2 to glutamylate putrescine shown in Figure 3 for the other polyamines? 

- Why did the authors use an alternative strategy and why this strategy gave such enormous errors bars in Figure 4?

Can the authors conclude anything about the potential  different substrate specificity of GlnA2 and GlnA3 based on the experiment reported in  Figure 4 ?

Figure 6

I guess that Sd stands for spermine and sp for spermidine. It would be nice to mention it in the figure legend. Furthermore I would suggest to use Spd rather than sd for spermidine.

I would analyze data of Figure 6  as follows:  in the GlnA2 mutant one sees the activity of GlnA3  and in the GlnA3 mutant one sees the activity of GlnA2.  This can give an idea of substrate specificity of each enzyme. So considering growth features (that could have been more quantitative…) spermine and spermidine would be better substrates for GlnA3 than for Gln2 whereas putrescine would be better substrate for GlnA2 than for GlnA3 and cadaverine is a bad substrate for both GlnA2 and GlnA3. These data are consistent with and thus support those of Figure 4, so it is surprising that the authors did not follow such a simple thinking that would greatly improve the clarity of their manuscript.

Figure 7

The logic behind Figure 7 is unclear. This figure should provide intracellular polyamine concentrations for the 3 or 4 polyamines (Put, Cad, Sp and  Spd) for the four strains (WT, and glnA2, A3 and A4 mutants) so there is a lot of missing information and this incomplete figure cannot be accepted as it is. Furthermore how do the authors explain the fast disappearance of Put in the glnA2 mutant?
They should at least comment this surprising result.

MINOR EDITORIAL CORRECTIONS

36 From our results, we can propose a complemented model of polyamine catabolism
in S. coelicolor, which involves the gamma-glutamylation pathway as well as other alternative utilization pathways.

93 All pathways aim to the neutralization of polyamines. However, some of these
pathways yield toxic intermediates or end products: gamma-aminobutyraldehyde appears in the aminotransferase (Fig. 1, AMTP), direct oxidation (Fig. 1, DOP) and spermine/spermidine dehydrogenase (Fig. 1, SPDP) pathways. Acetylspermidine occurring in the acetylation pathway (Fig. 1, ACP) may not
be metabolized further and may need to be excreted. In contrast, the gamma-
glutamylation pathway (Fig. 1, GGP) is suitable for further metabolization of polyamines as N-sources and it was demonstrated that in this pathway polyamines can be reduced to glutarate and succinate that enter the TCA cycle [4, 17]. The enzyme catalyzing the initial step of the gamma-glutamylation pathway is the gamma-glutamylpolyamine synthetase ligating polyamines with glutamate.

 109 The polyamine gamma-glutamylation pathway was also described in S. coelicolor.

114 These enzymes do not exhibit glutamine synthesis activity [19] that is achieved by two functional glutamine synthetases (GSs): GS type I – GlnA (SCO2198) and GS type II - GlnII (SCO2210) [20, 21, 22].

121 It was shown that in Mycobacterium…

  1. Results
    2.1. The sequence and the 3D model structure of GlnA2 S. coelicolor M145 differs from that of glutamine synthetases, but resembles that of glutamate-polyamine ligases 132
    Three genes encoding GlnA-like enzymes are present in the S. coelicolor genome.

138To the best of our knowledge there are no reports on the function of glnA2 in Actinomycetes. It is only known that glnA2 does not code for an enzyme with GS activity: the glnA2 gene cannot complement a double mutant of the glutamine synthetase genes glnA and glnII in S. coelicolor [1, 19]. In order to find indications on GlnA2 function, the amino acid sequence of GlnA2 was compared with glutamine synthetases. The sequence alignment of GS and GS-like proteins from S. coelicolor and other bacteria by BLASTp revealed 18 highly conserved amino acid residues. For instance, GlnASt 145
from Salmonella typhimurum and GlnASc from S. coelicolor contain identical conserved amino acids whereas GS-like proteins of S. coelicolor have multiple substitutions of conserved amino acids. GlnA2 has 32% identity / 48% similarity with GlnASc. In S. coelicolor, GlnA2 and GlnA3 lack some amino acids present in the active site of GlnA (Suppl. Fig. 1). Two of the 18 amino acid residues present in the active site of GS and important for their activity (Ser53, Asn26) are absent  in GlnA2, while other conserved residues D50, Tyr179, Glu327 and W397 are present  (Suppl. Fig. 1).
For further detailed in silico analysis of proteins, comparative 3D modelling of GlnA2 with the crystal structure of described glutamine synthetases was performed. The aim of the analysis of the GlnA2 3D model structure was to identify the localization of the amino acid residues present in GlnA but absent in GlnA2.

161 This protein template yielded the best GlnA2 3D model as evaluated using the QMEAN scoring function, the MolProbility score and a Ramachandran plot. The GlnA2 3D model consists of 6 sub-units organized in two rings, forming a dodecameric enzyme (Fig. 2, A). A structural comparison
of the GlnA2 model with the crystal structure of GSI from S. typhimurium (Fig. 2, B) revealed conserved binding sites for glutamate, ATP and metal ions Mg2+/Mn2+ in the active site of GlnA2. Moreover, it revealed that the amino acid substitutions in GlnA2 are 167
localized in a region corresponding to the ammonium binding site in GlnA:

171 GlnA3 is a closely related to GlnA2 that ligates glutamate with polyamines [1]. The comparison of its 3D model structure with that of GSI GlnA from S. typhimurium revealed substitutions of amino acid residues in the ammonium binding site:

176 This suggests the possibility of the binding of substrates other than ammonium,

198 The reaction mix for the in vitro reaction was containing the purified His-GlnA2 enzyme, glutamate, putrescine, ATP and Mg2+ and was incubated at 30 ̊C for 5 min.

223 …which are about fifty fold higher than the physiological ones for ammonium. In in vivo tests GlnA2 did not show any GS activity [1, 19]. The measured GS activity could presumably be attributed to

247 …the transcription levels of the glnA2 and glnA3 genes were similar and remained constant and were lower than that of glnA [30].

250 Total RNA was isolated from S. coelicolor M145 grown in defined Evans medium

Change or paragraph order:

Transcriptional analysis of glnA2 in the presence of polyamines revealed increased 269
expression levels of the glnA2 gene in medium with putrescine, cadaverine and 270
spermidine as the only N-source (Fig. 5, B). Such enhanced expression was previously 271
observed also for glnA3 in the presence of polyamines [1]. These results indicate that 272
putative substrates of the GlnA2 enzyme induce the expression of glnA2.

In order to confirm the absence of the influence of glutamine on the glnA2 expression, a
luciferase-based reporter gene analysis under different nitrogen conditions was performed. To do so, the luciferase encoding genes luxAB lacking a promoter were fused to the approximately 200 bp long promoter region of glnA2 and integrated into the genome of S. coelicolor M145. The luminescence was measured in samples incubated in defined media with sole nitrogen sources ammonium, nitrate, glutamate and glutamine. The assay demonstrated no expression of glnA2 in the presence of 5 mM or 50 mM glutamine as the only nitrogen source (Suppl. Fig. 4).
       In contrast, enhanced expression of glnA2 was observed under starvation conditions with low glucose and ammonium concentration (Fig. 5, C), but no expression was detected in the medium with high amounts of ammonium or glutamine (Fig. 5, A, C). This expression pattern is similar to that reported for glnA3 [1], glnA4 [23], glnA and glnII [32] under similar conditions.

283 In order to elucidate the role of GlnA2 in the physiology of the cell, a ΔglnA2 deletion mutant was generated. The glnA2 gene was interrupted by the apramycin resistance gene, which was introduced into the genome by double crossover using the flanking regions of the glnA2 gene. The deletion of glnA2 was confirmed by the successful genetic complementation of the mutant with the wild type glnA2 gene. For subsequent phenotypical analysis, the glnA2 mutant was tested for its ability to utilize different nitrogen sources. The ΔglnA3 [19] and ΔglnA4 [1, 23] mutants as well as the parental strain S. coelicolor M145 were also included in this analysis.

291  First, to investigate the potential toxicity of high polyamine concentrations on S. coelicolor,

340 In order to obtain further information on the role of GlnA2 in polyamine utilization as well as on the origin of polyamine toxicity…

357 The glnA3 deletion was previously reported to result in the intracellular accumulation of putrescine, cadaverine and spermidine [1] …

388  …indicating that the acetylated version of GlnR binds better to the glnA2 promoter region than the non-acetylated version of GlnR.

430 Although GlnA2 and GlnA3 are both involved  in polyamine metabolism, the transcriptional regulation of their encoding genes is complex and controlled by different regulators.

450… pointing out on the ability of GlnA2, alone, to utilize polyamines.

463 Other reports describe the role of GlnA2 in Halobacillus halophilus. In this halophilic
bacterium, the transcription of glnA2 is influenced by NaCl and GlnA2 can act as a GS participating in the synthesis of glutamine as compatible solute [36]. In contrast, in S. coelicolor salt concentrations up to 1 M NaCl had no influence on glnA2 transcription.

492 In the next utilization step polyamines might be converted to gamma- aminobutyraldehydes (Fig. 9). The production of intermediate products resulting from polyamine catabolism  such as aminobutyraldehyde can be toxic for the cell [17]. Our results in combination with a previous report [1] suggest the conversion of these intermediates to gamma-aminobutyric acid (GABA) at different stages of polyamine utilization in S. coelicolor (Fig. 9), are similar to those of P. aeruginosa and E. coli.

502 The enhanced expression of sco5676 in presence of polyamines [1] as well as the interaction of the regulator of polyamine genes EpuRI with the promoter region of SCO5676 reported  in this work suggest a role of SCO5676 in the conversion of GABA to semialdehydes (Fig. 9).

510 The utilization of putrescine can occur not only via the gamma-glutamylation pathway, but also via the aminotransferase pathway or direct oxidation pathway, as in P. aeruginosa. A putative amidotransferase (SCO5655) was identified in silico and is a predicted to be an homolog of the putrescine amidotransferase (PatA) from E. coli.

521 Interestingly, the gene sco5671 was expressed in the presence of spermidine, but not in the

527 In summary, Streptomyces apparently possess a complex network of enzymes and regulators to metabolize different polyamines …

530 The complexity of the network is also reflected by the occurrence of different metabolic pathways and numerous regulators contributing to the  transcriptional control of the various metabolic genes. However a  central role seemed to be played by the two gamma-glutamylpolyamine synthetases GlnA2 and GlnA3 that initiate polyamines metabolization.

Author Response

(The authors gave the same response as above.)
